# Microbiome Profile of the Mediterranean Mussel (*Mytilus galloprovincialis*) from Northern Aegean Sea (Greece) Culture Areas, Based on a *16S rRNA* Next Generation Sequencing Approach

Konstantinos Schoinas [1,†], Vasiliki Konstantou [1,†], Emmanouela Bompou [1], George Floros [2], Dimitrios Chatziplis [3], Anastasia Imsiridou [1] and Dimitrios Loukovitis [4,*]

1   Department of Food Science and Technology, School of Geosciences, International Hellenic University, 57400 Thessaloniki, Greece
2   Directorate of Veterinary Center of Thessaloniki, Ministry of Rural Development and Food, 54627 Thessaloniki, Greece
3   Laboratory of Agrobiotechnology and Inspection of Agricultural Products, Department of Agriculture, School of Geosciences, International Hellenic University, 57400 Thessaloniki, Greece
4   Department of Fisheries and Aquaculture, School of Agricultural Sciences, University of Patras, 30200 Messolonghi, Greece
*   Correspondence: dloukovi@upatras.gr
†   These authors contributed equally to this work.

**Abstract:** Mediterranean mussels (*Mytilus galloprovincialis*), due to their nutritional mechanisms which involve filtering huge amounts of water, are affected by seawater pollution and can host microbial diversity of environmental origin, as well as pathogenic bacteria that must be constantly monitored. Herein, we applied a Next Generation Sequencing (NGS) metabarcoding approach in order to study the *M. galloprovincialis* microbiota. Collection of samples was conducted during winter and summer months from various mussel farm zones located in specific farm regions in the Thermaikos gulf, the northern Aegean Sea, Greece. A microbiological test was performed for the enumeration of *Escherichia coli* and the presence of *Salmonella sp*. DNA extraction and amplification of the whole bacterial *16S rRNA* gene, followed by NGS amplicon sequencing and taxonomic classification, were carried out. Statistically significant differences ($p < 0.05$) in the abundance of the most dominant bacterial phyla, families and genera between winter and summer time periods, regions, as well as zones within each region of sampling, were evaluated with *z*-score computation. According to the obtained results, the most prevalent taxa at the genus level were *Mycoplasma* (12.2%), *Anaplasma* (5.8%), *Ruegeria* (5.2%) and *Mariniblastus* (2.1%). Significant differences in the abundance of the most dominant genera were found at all levels of comparison (seasons, regions and zones within each region), highlighting the dynamic character of microorganisms, which might be affected by microenvironmental, temporal and spatial changes. The present research contributes to the characterization of *M. galloprovincialis* microbiome in areas that have not been studied previously, setting the baseline for future, more thorough investigations of the specific bivalve species and its bacterial profile in the above geographic regions.

**Keywords:** Mediterranean mussel; *Mytilus galloprovincialis*; *16S rRNA*; metabarcoding; microbiome; NGS

## 1. Introduction

The marine ecosystem consists of an immense number of organisms, with *Mollusca* being one of the most considerable phyla. A significant taxonomic unit in this group is the class of *Bivalvia*, with the presence of a double shell as its main feature, which contains, among others, the genus *Mytilus*. Globally, the vast majority of mussel farms cultivate the species *Mytilus galloprovincialis* and *M. edulis* [1]. More specifically, *M. galloprovincialis*

(Mediterranean mussel) displays a key role in the marine ecosystem and has a relevant economic value, as a species of interest, in many coastal areas (e.g., Mediterranean Sea, Black Sea, Atlantic coasts) [2,3].

Generally, mussels as filter-feeding organisms are capable of filtering high volume of seawater (7.5 L/h, 25 °C) through their gills [4]. In this way, mussels percolate small particles (3–5 μm) and microorganisms, such as planktons, fulfilling their nutritional needs. Consequently, they accumulate various substances, such as heavy metals, microplastics and antibiotics, as well as bacteria, both pathogenic and non-pathogenic, in their tissues [2,5]. Therefore, mussels contribute to reducing the eutrophication phenomenon and are usually utilized as bioindicators to monitor the pollution of coastal areas [6]. The accumulated bacteria which compose the mussel's microbiome have an active role in its life. The bacterial communities provide probiotic functions, such as improved digestion, immunological regulation and defense against pathogens. Reciprocally, the host provides a steady substrate and a constant nutrient supply [3,7]. The composition and diversity of mussel's microbiome is impacted by two main factors, environmental fluctuations including temperature, pH, salinity, oxygen and anthropogenic interferences including urban, agricultural and industrial wastewater [8,9].

In Greece, the Thermaikos Gulf accommodates the majority of mussel farms [10]. It is located in the northwestern part of the Aegean Sea and is a river-fed and semi-enclosed bay with an average depth of 20–60 m. This gulf is mainly affected by four rivers, primarily the Axios River and secondarily the Aliakmonas River, the Loudias River and the Gallikos River [11]. The Axios River has a high rate of outflow during spring and a low rate in late summer [12]. All of these rivers contribute to the deposition of fresh water, nutrients and the accumulation of pollutants [11]. The Thermaikos Gulf is susceptible to extended anthropogenic pressure, mainly in its northern sector, and several meteorological conditions. Thessaloniki, the second largest city in Greece situated in the northern part of the Thermaikos Gulf, has a crucial impact on the coastal ecosystem due to the presence of the port, the touristic activities, the sewage water and the industrial and rural effluents [12]. The climate of the gulf is characterized as Mediterranean, with warm and dry summers, mild winters and temperatures ranging from 1.5 °C–31.7 °C (based on monthly measurements from 1959 to 2010). Up to now, the Thermaikos Gulf and other mussel-crowded sea areas in Greece have not been sufficiently studied regarding the mussel's microbiome.

The microbiome of the mussel can be identified with many methods, with one of them being the culture-dependent bacteriological analysis. The plate count method, though, is prone to several limitations due to the risk of contamination, time and resource consumption, demand of experienced analyst, questionable reproducibility and the reliance on phenotypic biochemical characterization which can be easily altered [13,14]. Moreover, heterotrophic plate counts could be critically affected by temperature [15]. Besides that, the main drawback of this method is that not all bacteria can be cultured; in fact, only 1% of the prokaryotes in most environments can be cultivated in isolation [16,17].

Throughout the last decades, the development of molecular techniques, particularly Next Generation Sequencing (NGS) approaches, enabled a more accurate and reliable analysis of a host's microbiome. Thus, these techniques were used for the identification of the mussel's microbiome in previous studies, which were mainly focused on the Second Generation Sequencing (SGS) methodologies. For instance, Bozcal and Dagdeviren [2] and Musella et al. [3] applied the genotyping technology of Illumina Inc. (San Diego, CA, USA) to study the microbiome of *M. galloprovincialis*, targeting the V3–V4 hypervariable region of the *16S rRNA* gene. Likewise, Wathsala et al. [9] conducted another study aiming at the microbiome of the digestive gland. In addition, Auguste et al. [18] utilized the ION Torrent sequencing technique (Thermo Fisher Scientific Inc., Waltham, MA, USA) to determine the mussel's microbiome biodiversity through the V4 hypervariable region. Furthermore, Balbi et al. [19], using the same technology and *16S rRNA* gene region, investigated the mussel's microbiome emphasizing on the early stages of the mussel's life. Another SGS

methodology is the 454 pyrosequencing (Roche Holding AG, Basel, Switzerland), which Vezzuli et al. [20] applied to the gene's V6 hypervariable region in order to study the microbiome profile of the mussel's hemolymph and digestive gland.

The *16S rRNA gene* is, undoubtedly, the main and most used gene for prokaryotes identification. The significant intraspecific differences of this gene can enable the identification of bacteria not only at the genus level, but even at the species level [21]. The hypervariable regions targeted in the aforementioned studies represent, however, only a small fraction of the gene, which corresponds to up to 460 bp. On the other hand, the latest NGS approaches, i.e., Third Generation Sequencing (TGS), are capable of analyzing the entire sequence of the *16S rRNA* gene, being approximately 1500 bp long, that includes the V1-V9 hypervariable regions. In consequence, the long-read sequencing techniques can be much more informative in metabarcoding studies of microorganisms, providing the opportunity for higher resolution at lower taxonomic levels.

To the best of our knowledge, studies that are related to the mussel's microbiota and how it might be affected by spatial/temporal changes, environmental and anthropogenic factors, are still limited. Therefore, the aim of this study was to analyze, for the first time, the microbiome profile of the Mediterranean mussel from culture areas of the northern Aegean Sea with respect to different time periods and sampling zones based on a Third Generation Amplicon Sequencing approach of the whole *16S rRNA* gene. Characterization of the microbial composition may contribute significantly to the elucidation of the role and function of microbiota in bivalves, providing essential information for further studies.

## 2. Materials and Methods

### 2.1. Site Information and Sampling

Sample collection was conducted twice during the winter (December–February) and summer (July, August) months of 2021–2022 in mussel farms of the northern Aegean Sea (Thermaikos Gulf). The mussels were collected from different zones (corresponding to different mussel farms) of Chalastra and Makrigialos regions by a professional veterinarian, according to the suggested sample collection protocol for microbiological tests. The sampling was carried out at 3–8 m depth and the temperature, along with the salinity, were measured at the surface of each sampling location (Table 1). Within a few hours, the mussels were transferred to the Laboratory of the Directorate of Veterinary Center of Thessaloniki (Thessaloniki, Greece) for the initial analysis and, subsequently, to the Laboratory of Agrobiotechnology and Inspection of Agricultural Products of the International Hellenic University (Thessaloniki, Greece), where they were stored for further analysis. Throughout the transportation process, the temperature of the samples was preserved at $\leq 4$ °C, maintaining the cold chain conditions with the use of polystyrene containers and ice packs.

### 2.2. Sample Preparation

Sample preparation, along with the microbiological tests, were conducted at the Laboratory of the Directorate of Veterinary Center of Thessaloniki. For each sample, 10–12 mussels were selected from each zone in alive conditions, in adult stage and of similar size (6.5–7.5 cm). Following the sample selection, the mussels were opened with a sterile knife and the whole content (digestive gland, gills, foot, mantle, liquid) was placed in a stomacher bag. Then, a two-minute homogenization process was performed using the BagMixer Stomacher device (Interscience, London, UK). Finally, the homogenized content was transferred to a 10 mL sterile syringe and the ready-to-analyze samples were stored at −80 °C. All samples were examined for the enumeration of *E. coli* with the Most Probable Number (MPN) method, according to ISO 16649-3:2005, and for the presence of *Salmonella sp.* with the RVS broth and XLD agar isolation method, according to ISO 6579-1:2017. In total, 15 samples were prepared and analyzed for the purpose of the present study (Table 1).

**Table 1.** Number of mussels used for each sample (total number of mussels for all samples = 176), sampling zone, collection period and characteristics of each region (the water surface temperature and salinity data were obtained from poseidon.hcmr.gr, accessed on 20 September 2022).

| Sample | Number of Mussels | Zone | Region | Month | Collection Depth (m) | Water Surface Temperature (°C) | Salinity (psu) |
|---|---|---|---|---|---|---|---|
| $M_1$ | 10 | $Z1_M$ | Makrigialos | December | 3–4 | 15.7 | 37–38 |
| $M_2$ | 12 | $Z2_M$ | | | | | |
| $M_3$ | 12 | $Z1_M$ | Makrigialos | January | 3–4 | 14.0 | 37–38 |
| $M_4$ | 12 | $Z2_M$ | | | | | |
| $M_5$ | 12 | $Z1_M$ | Makrigialos | February | 3–4 | 11.7 | 37–38 |
| $M_6$ | 12 | $Z2_M$ | | | | | |
| $M_7$ | 11 | $Z1_C$ | Chalastra | February | 4–8 | 11.0 | 37–38 |
| $M_8$ | 12 | $Z2_C$ | | | | | |
| $M_9$ | 12 | $Z3_C$ | | | | | |
| $M_{10}$ | 12 | $Z1_C$ | Chalastra | February | 4–8 | 12.1 | 37–38 |
| $M_{11}$ | 12 | $Z2_C$ | | | | | |
| $M_{13}$ | 12 | $Z1_C$ | Chalastra | July | 4–8 | 26.0 | 36–37 |
| $M_{14}$ | 11 | $Z2_C$ | | | | | |
| $M_{15}$ | 12 | $Z3_C$ | | | | | |
| $M_{16}$ | 12 | $Z2_M$ | Makrigialos | August | 3–4 | 27.5 | 36–37 |

### 2.3. DNA Isolation

Total bacterial DNA was extracted after thawing (to room temperature) the homogenized samples, using the DNeasy PowerFood Microbial Kit (Qiagen, Hilden, German) and according to the manufacturer's protocol with the following minor modifications. Each centrifugation process was adjusted to 14,000× *g* and an additional step of incubation at 65 °C in a water bath for 15 min was performed, after the resuspension of the microbial pellet to the MBL Solution. The centrifugation step for ethanol removal was repeated twice in order to limit the ethanol residuals. An extra incubation step at room temperature for four minutes was added after the elution step and the following centrifugation was performed twice. The second one was done after re-adding the flow-through of the first centrifugation into the filter column. With the completion of DNA extraction, the quantity and quality of all DNA samples were checked in an agarose gel through electrophoresis.

### 2.4. 16S rRNA Gene Library Preparation

The whole *16S rRNA* gene was amplified with the use of 16S Barcoding Kit 1–24 (SQK-16S024—Oxford Nanopore Technologies plc., Oxford Science Park, Oxford, UK) and the LongAmp® Hot Start Taq 2X Master Mix (New England Biolabs Inc., Ipswich, MA, USA), following the associated protocols with minor adjustments. The primers 27F (5′-AGAGTTTGATCCTGGCTCAG-3′) and 1492R (5′-GGTTACCTTGTTACGACTT-3′) were used, targeting the V1–V9 hypervariable regions with an amplicon length of ~1500 bp. The primers are attached to specific DNA nucleotides, serving as barcodes, which enables sample pooling at later stages. The PCR method was carried out according to the following program: 95 °C for 3 min as an initial denaturation step, denaturation at 95 °C for 30 s, annealing at 55 °C for 1 min, elongation at 65 °C for 2.5 min and 10 min at 65 °C as a final elongation step. The denaturation, annealing and elongation steps were repeated for 40 cycles. After that, the obtained amplicons were subjected to agarose gel electrophoresis for the confirmation of the *16S rRNA* gene amplification. Finally, the purity and concentration of the PCR products were determined with a Q3000 UV-spectrometer (Quawell, San Jose, CA, USA).

### 2.5. Library Purification and Amplicon Sequencing

The barcoded PCR products were pooled to equimolar quantities based on spectrophotometry results and the pooled sample was purified with MicroCLEAN reagent (Clent Life Science, Stourbridge, UK). The purified pooled library was checked again with spectropho-

tometry for its purity and quantity of DNA and the final concentration was adjusted to ~100 ng/µL by diluting with ultra-pure water (molecular biology grade).

As the primers used in the PCR are specific primers with rapid attachment chemistry, 1 µL of the pooled library was mixed with 1 ul of rapid 1D sequencing adapter from the 16S Barcoding Kit 1–24 and the library solution was properly prepared (by adding sequencing buffer, loading beads etc.) and loaded into a flowcell (version R.9.4.1, Oxford Nanopore Technologies plc.) placed in a MinION-Mk1B sequencing device (Oxford Nanopore Technologies plc.). The sequencing run parameters were configured by MinKNOW software (Oxford Nanopore Technologies plc.).

### 2.6. Bioinformatic and Statistical Analysis

Raw data (FAST5 files) were basecalled with algorithms implemented in GUPPY software (Oxford Nanopore Technologies plc.), where reads were demultiplexed according to the used barcodes. Clean *16S rRNA* sequences were obtained after trimming of barcodes, adapter and primer sequences and were, subsequently, subjected to EPI2ME Fastq 16S cloud-based bioinformatics workflow (Oxford Nanopore Technologies plc.) for taxonomic classification, setting an identity threshold of 90%. The specific workflow facilitates the classification of *16S rRNA* sequences down to the genus and, in some cases, to the species level, using the BLAST algorithm against a 16S ribosomal RNA database (bacterial and archaeal strains, NCBI—National Center for Biotechnology Information, Bethesda, MD, USA). Prior to taxonomic classification, all reads having a length below 1000 and above 2000 bp were excluded and, additionally, a quality score (Q-score) threshold of 12 was applied. Furthermore, an alpha rarefaction analysis was performed using the MetONTIIME statistical package [22,23] in order to examine the association between the sequencing depth and the richness of the bacterial community under study. Alpha diversity at the genus level was assessed using two distinct metrics, Shannon diversity index [24] and observed features. Finally, based on the classification results, pie charts and bar plots were built using Microsoft Office Excel (Redmond, WA, USA). Statistically significant differences ($p < 0.05$) in the abundance of the most dominant bacterial phyla, families and genera between winter and summer time periods, regions, as well as zones within each region of sampling, were evaluated with $z$-score computation. Pairwise comparisons were possible only between the same phyla, families and genera, e.g., percentage of *Mycoplasma* genus in Chalastra region with the percentage of the same genus in Makrigialos region during winter, difference in the abundance of *Ruegeria* genus between Z1 and Z2 zones of Chalastra region during summer season, comparison of the presence of *Mycoplasma* genus between winter and summer time periods in Makrigialos region etc. The statistical analysis was carried out with Minitab v21.1.0 software (https://www.minitab.com/en-us/, accessed on 10 October 2022).

### 3. Results

#### 3.1. Microbiological Analysis

The microbiological analysis that was conducted for the enumeration of *E. coli* with the Most Probable Number (MPN) method revealed a low presence of *E. coli*, <230 MPN/100 g for all the 15 samples, according to the 2015/2285 regulation. Therefore, Makrigialos and Chalastra are classified as class A (intended for direct human consumption) production areas, based on the Codex Alimentarius criterion for *E. coli*. The ISO 6579-1:2017 method confirmed the absence of *Salmonella sp.* using 25 g from each analyzed sample.

#### 3.2. Metagenomic Analysis

DNA extraction, *16S rRNA* gene amplification and amplicon sequencing were successful for all 15 samples. Regarding taxonomic classification results, a total of 671,794 classified reads were obtained from EPI2ME Fastq 16S cloud-based bioinformatics workflow. The distribution of the assigned reads to each barcoded sample ranged from 30,000 to 95,000 reads with an average of 45,000 reads (per barcode), except for sample 16 (BC16), which had

approximately 20,000 reads. The 15 samples with their corresponding barcodes are shown in Table 2.

**Table 2.** The analyzed samples with their corresponding barcodes.

| Sample | $M_1$ | $M_2$ | $M_3$ | $M_4$ | $M_5$ | $M_6$ | $M_7$ | $M_8$ |
|---|---|---|---|---|---|---|---|---|
| Barcode | BC01 | BC02 | BC03 | BC04 | BC07 | BC08 | BC05 | BC06 |
| Sample | $M_9$ | $M_{10}$ | $M_{11}$ | $M_{13}$ | $M_{14}$ | $M_{15}$ | $M_{16}$ | |
| Barcode | BC09 | BC10 | BC11 | BC13 | BC14 | BC15 | BC16 | |

Alpha rarefaction analysis was carried out using two different alpha diversity metrics: observed features and Shannon diversity index. As shown in Figure 1a,b, all samples almost reached the plateau phase according to the rarefaction curves of the bacterial population at the genus taxonomic level. This implies an adequate sequencing depth and any additional increase in the number of *16S rRNA* sequences would have a minor effect in the number of genera revealed.

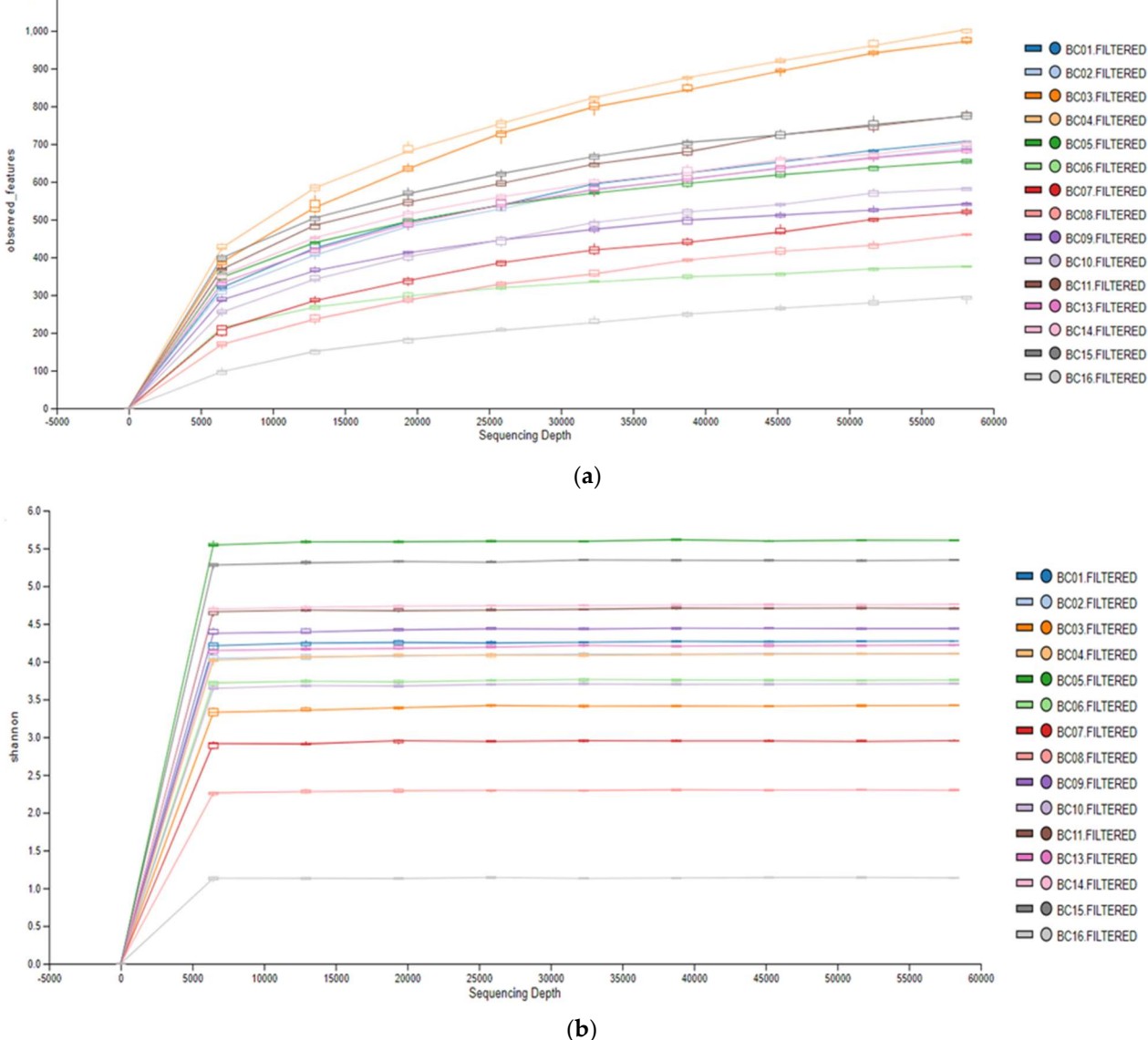

(a)

(b)

**Figure 1.** Alpha rarefaction curves of all samples at the genus level: (**a**) the sequencing depth in relation to the observed features and (**b**) the sequencing depth in relation to Shannon index.

Metabarcoding analysis revealed the mussels' bacterial diversity in the examined areas and the identified taxonomic units were compared in terms of zone, region and season. The relative abundance of the taxonomic units was calculated using only the classified reads and the overall composition of the *M. galloprovincialis* microbiome is presented in Figure 2 (Figure 2a–e). The dominant taxa at the phylum level were Proteobacteria (53.8%), Tenericutes (13.1%) and Bacteroidetes (8.2%). At the class level, the most abundant taxon was Alphaproteobacteria (32.1%), followed by Gammaproteobacteria (17.4%) and Mollicutes (13.1%). The dominant taxa at the family level were Mycoplasmataceae (13.1%), Rhodobacteraceae (11.6%) and Anaplasmataceae (5.9%). The most prevalent taxa at the genus level were *Mycoplasma* (12.2%), *Anaplasma* (5.8%) and *Ruegeria* (5.2%). At the species level, *Mycoplasma gypis* (3.9%), *Anaplasma phagocytophilum* (3.8%), *Mycoplasma falconis* (3.2%), *Ruegeria atlantica* (2.8%), *Mycoplasma gateae* (2.4%), *Mariniblastus fucicola* (2.1%) and *Anaplasma odocoilei* (2.0%) were the most represented taxa.

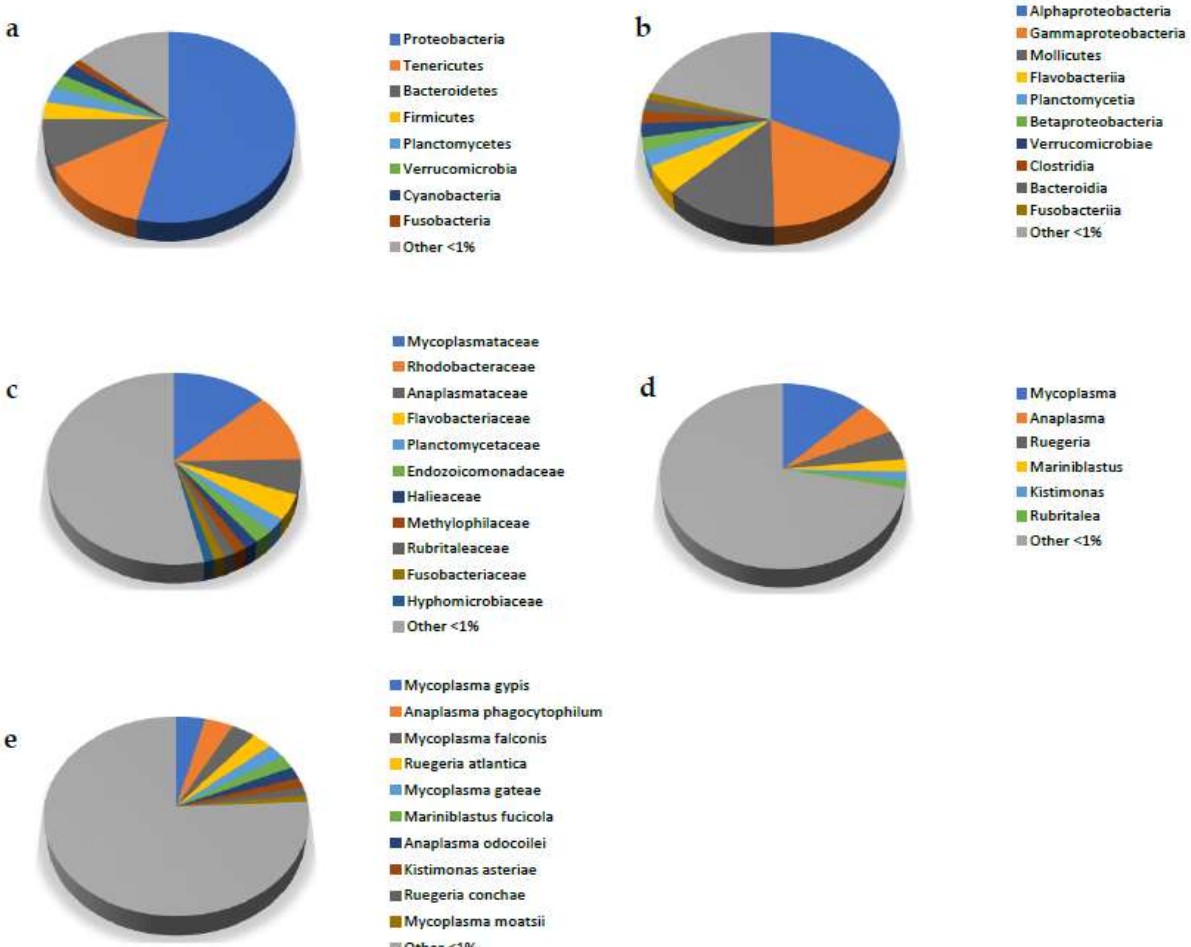

**Figure 2.** The whole M. galloprovincialis microbiome at the (**a**) phylum, (**b**) class, (**c**) family, (**d**) genus and (**e**) species levels. Pie charts were obtained using only the classified reads of the 15 samples. Only bacteria with relative abundance ≥1% were included.

The taxonomy results were further utilized in order to evaluate the dynamics of the bacterial communities under study. Thus, pairwise comparisons (and their corresponding statistical significance) of bacterial diversity between Chalastra/Makrigialos sampling regions and winter/summer time periods, at the phylum and family taxonomic levels, were carried out (Figure S1). Furthermore, the bacterial diversity between the three zones of Chalastra region was examined, both in winter and summer months. The same comparison was conducted between the two zones of Makrigialos region in the winter months and the results are presented in Figure 3a–c). As shown in Figure 3a, during winter in Chalastra

region, the most abundant bacterial genera in zones Z1 and Z3 were *Mycoplasma* (9.8% and 19.6%, respectively), *Mariniblastus* (2.8% and 5.6%, respectively) and *Rubritalea* (1.9% and 4.4%, respectively), while, in zone Z2 *Vibrio* (2.6%) held the third position, after *Mycoplasma* (7.3%) and *Mariniblastus* (3.7%). Z-score analysis revealed that all comparisons between the zones (Z1–Z2, Z1–Z3 and Z2–Z3) were statistically ($p < 0.05$) significant (different), with the exception of *Rubritalea* between Z1 and Z2 ($p = 0.2113$). During summer season in the region of Chalastra (Figure 3b), the most dominant bacterial genera of zones Z1, Z2 and Z3 were *Mycoplasma* (37.9%, 13.9% and 2.2%, respectively), *Ruegeria* (5.4%, 8.2% and 10.9%, respectively) and *Kistomonas* (5.1%, 1.8% and 2.5%, respectively). Z-score analysis showed that all comparisons between the zones (Z1–Z2, Z1–Z3 and Z2–Z3) were significantly different ($p < 0.05$). Furthermore, during winter in Makrigialos region (Figure 3c), the most represented bacterial genera in zones Z1 and Z2 were *Anaplasma* (21.6% and 20.1%, respectively), *Mycoplasma* (11.8% and 3.0%, respectively), *Polaribacter* (2.5% and 3.7%, respectively) and *Ruegeria* (2.3% and 2.7%, respectively). Z-score analysis revealed in this case also that all comparisons between the zones (Z1–Z2) were statistically different at the 5% level of significance.

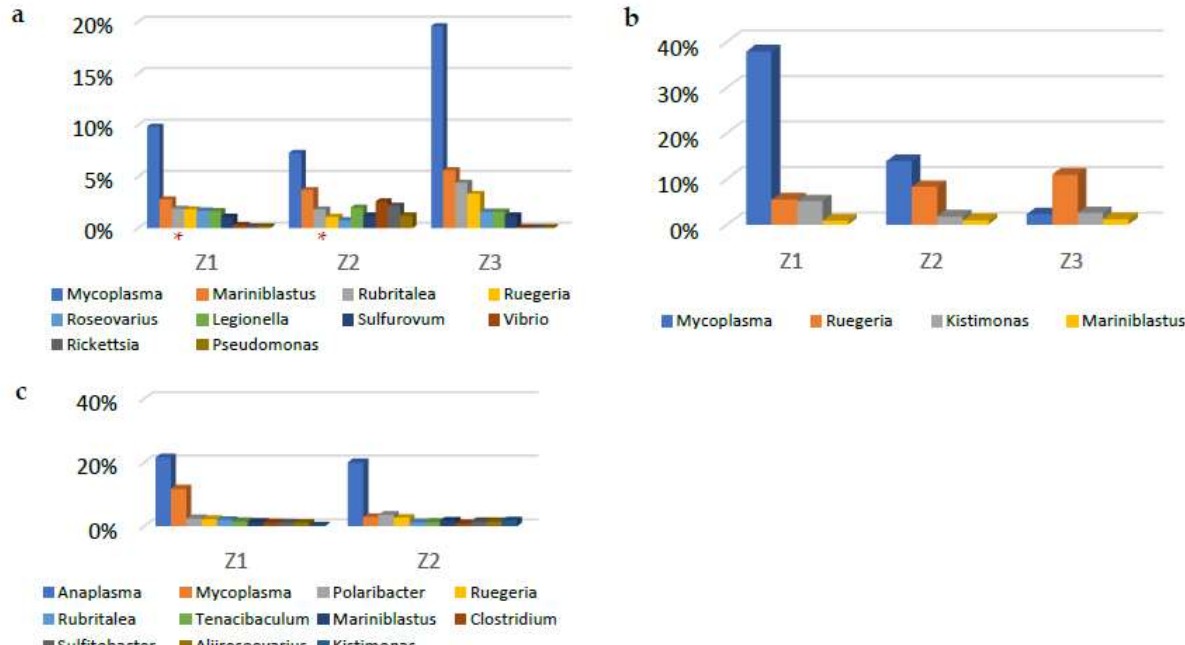

**Figure 3.** The most abundant bacteria at the genus level in the different zones of Chalastra and Makrigialos regions. (**a**) Chalastra winter, (**b**) Chalastra summer and (**c**) Makrigialos winter. Statistically non-significant pairwise comparisons are marked with an asterisk.

The results of the statistical comparisons between Chalastra and Makrigialos regions, both in winter and summer seasons, are presented in Figure 4a,b. In winter, the most abundant bacterial genera in Chalastra were *Mycoplasma* (11.3%), *Mariniblastus* (3.9%) and *Rubritalea* (2.5%), whereas in Makrigialos, *Anaplasma* (20.9%), *Mycoplasma* (7.6%) and *Polaribacter* (3.1%) dominated. All comparisons between the two regions in winter were found to be statistically different ($p < 0.05$), based on z-score analysis results. In summer time period, the most dominant bacterial genera in Chalastra were *Mycoplasma* (16.4%), *Ruegeria* (8.4%) and *Kistimonas* (3.0%), while in Makrigialos, *Ruegeria* (42.5%), *Mycoplasma* (15.9%) and *Clostridium* (3.7%) were represented the most. In this case, z-score tests showed that all comparisons between the two regions in summer were significantly different ($p < 0.05$), except for *Mycoplasma* genus ($p = 0.1499$).

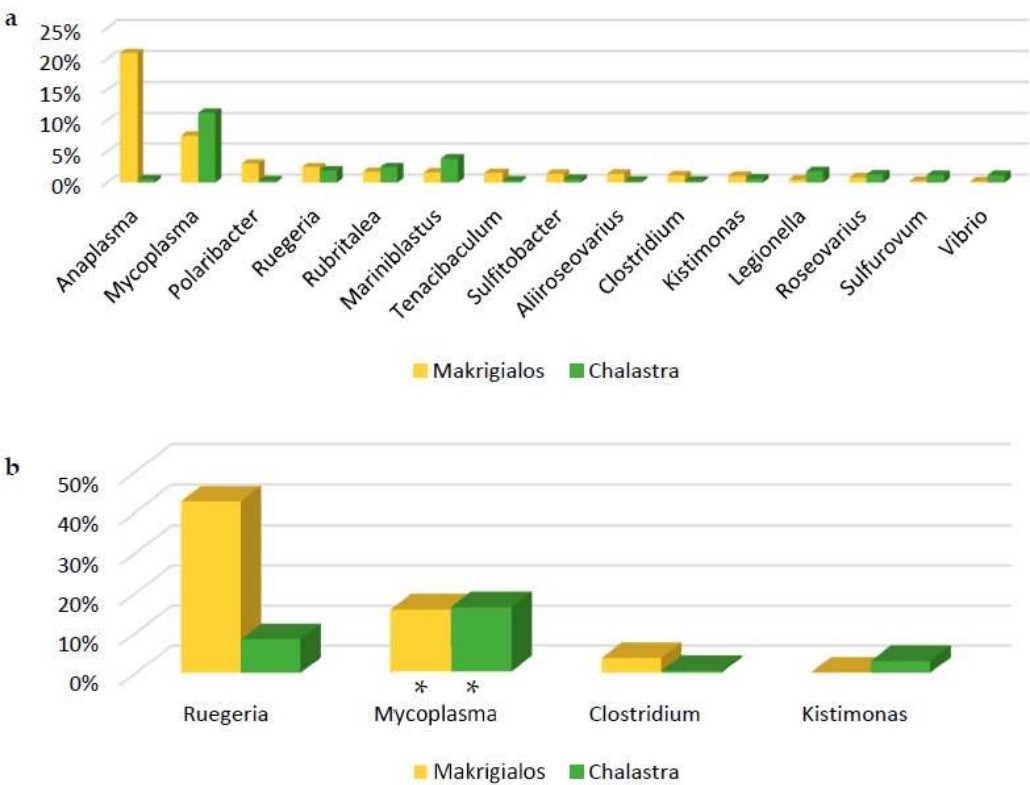

**Figure 4.** Comparison of the most abundant bacterial genera between (**a**) Chalastra-Makrigialos in winter and (**b**) Chalastra-Makrigialos in summer. Statistically non-significant pairwise comparisons are marked with an asterisk.

In terms of seasonal variability within each region, the results are presented in Figure 5a,b. Regarding Chalastra during the winter season, the most abundant bacterial genera were *Mycoplasma* (11.3%), *Mariniblastus* (3.9%), *Rubritalea* (2.5%) and *Ruegeria* (1.9%), whereas in summer time period, *Mycoplasma* (16.4%), *Ruegeria* (8.4%), *Kistimonas* (3.0%) and *Mariniblastus* (1.0%) were the most prevalent. *Z*-score analysis pointed out that all comparisons between the two aforementioned seasons in Chalastra were statistically different ($p < 0.05$). In Makrigialos region, the most dominant bacterial genera in winter were *Anaplasma* (20.9%), *Mycoplasma* (7.6%), *Polaribacter* (3.1%) and *Ruegeria* (2.5%), while in summer season *Ruegeria* (42.5%), *Mycoplasma* (15.9%) and *Clostridium* (3.7%) were the most represented. Again, all statistical comparisons carried out with *z*-score tests were different at the 5% level of significance. A remarkable difference in this specific region was the high prevalence of *Anaplasma* genus during winter and its total absence in summer season.

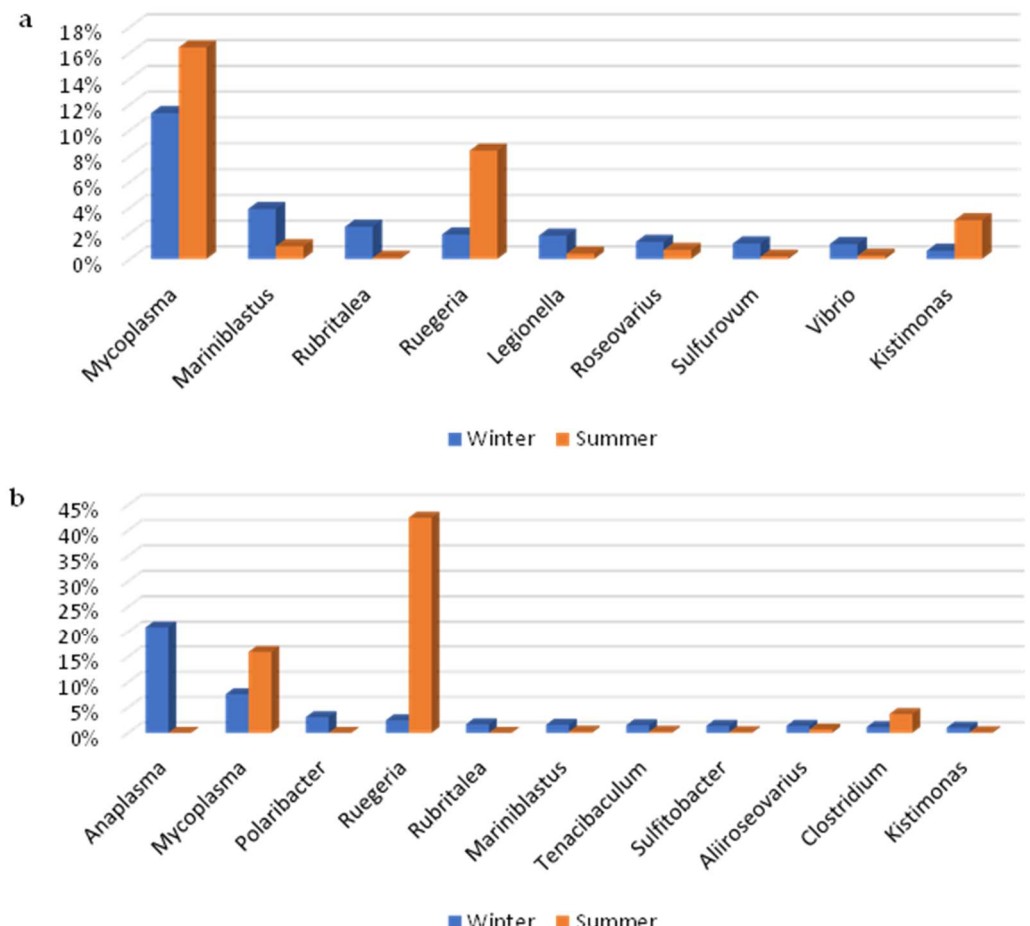

**Figure 5.** Comparison of the most abundant bacterial genera between (**a**) winter-summer in Chalastra region and (**b**) winter-summer in Makrigialos region.

## 4. Discussion

In the present study, we aimed to frame the *M. galloprovincialis* microbial profile with the most advanced technologies up to now, by exploring the most mussel-productive geographical sites in Greece. Previous studies were carried out in the Thermaikos gulf with the use of culture-dependent bacteriological analysis and/or first generation sequencing techniques (i.e., Sanger sequencing), revealing though a much lower bacterial diversity. More specific, an investigation conducted from Kalaitzidou et al. [25], in which water samples were collected from the respective gulf, showed *Sulfitobacter*, *Halomonas*, *Planococcus* and *Synechocystis* as the main halophilic bacterial genera. An additional study in the Thermaikos gulf, focusing on the microbiome in sediment samples, detected Beta-Gamma-Proteobacteria (31.7%), Deltaproteobacteria (23.8%), Acidobacteria (1.1%), Planctomycetales (7.9%), Bacteroidetes (4.8%) and Alphaproteobacteria (3.2%) as the most represented taxonomic groups [26].

Studies involving NGS techniques such as Illumina-based genotyping of V3-V4 hypervariable regions of the *16S rRNA* gene mentioned various results depending on the taxonomic level. For instance, Bozcal and Dagdeviren [2] reported as dominant phyla the Proteobacteria, Firmicutes, Bacteria unclassified, Bacteroidetes and Actinobacteria. At the genus level, *Arcobacter*, *Clostridium*, *Vibrio*, *Pseudomonas* and *Mycoplasma* were identified, matching our results, though in lower abundance, with the exception of *Mycoplasma* genus. On the contrary, some other of their classified genera, such as *Aeromonas*, *Escherichia*, *Dokdonella* and *Morganella,* were absent in the present study. Similarly, Musella et al. [3] presented Proteobacteria, Firmicutes and Bacteroidetes as the most abundant phyla. Another study [18] also revealed the above-mentioned phyla and determined Gammaproteobacteria

and Bacteroidia as the main classes. In the same study, the dominant taxonomic units at the genus level were *Shewanella*, *Vibrio* and *Mycoplasma*.

Herein, the obtained results showed that *Mycoplasma* (12.2%), *Anaplasma* (5.8%), *Ruegeria* (5.2%) and *Mariniblastus* (2.1%) genera prevailed in the mussel's microbiome. The genus *Mycoplasma* was present in all samples regardless of region and season. Several studies have also reported similar abundance in bivalves. King et al. [27], Lokmer et al. [28] and Arfken et al. [29] mentioned high predominance of *Mycoplasma* in different tissues of oyster samples, which might indicate that this genus has probably developed a symbiotic mechanism with various bivalves and is, therefore, part of their core microbiome. On the contrary, according to Lattos et al. [30], the genus *Mycoplasma* has been generally reported as a potential pathogen in bivalves in elevated temperatures. Furthermore, *Anaplasma* genus (5.8%), as shown in Figure 2d, was the second most abundant genus in this study and a noteworthy difference in its relative abundance was observed between Makrigialos and Chalastra regions during winter (20.9% and 0.5%, respectively). Cano et al. [31] carried out research with diverse molluskan samples and found *Anaplasma* genus as one of the most represented genera, associating it with intracellular parasite-like behavior.

According to Arahal et al. [32], the genus *Ruegeria* belongs to the Roseobacter group, containing mostly marine members of the family Rhodobacteraceae. As a marine bacterium, it has been isolated from marine environments and has been characterized as a symbiont of marine invertebrates with mesophilic growth conditions [32,33]. The above information is in concordance with our findings presented in Figure 5b, where *Ruegeria* genus possessed a high percentage (42.5%) during summer in Makrigialos region, while in winter it was significantly lower (2.5%). Regarding *Mariniblastus* genus, the available resources for bivalvia are still limited. Previous studies from Lage et al. [34] and Faria et al. [35] have associated it with the complex biofilm of macroalgae in seawater. Subsequently, the filter-feeding mechanism of *M. galloprovincialis* along with the presence of *Mariniblastus* in the marine environment make their coexistence rather reasonable.

Public health concerning bacteria, i.e., *E. coli* and *Salmonella* spp., are usually detected with conventional methods such as MPN and biochemical techniques, which may, however, contain a form of bias. Mannas et al. [36] performed MPN for the enumeration of *E. coli* and biochemical identification for *Salmonella* spp. in *M. galloprovincialis* samples collected along the Moroccan Atlantic coast and detected both pathogens. Bozcal and Dagdeviren [2] conducted both MPN and metagenomic analysis for the detection of *E. coli* in mussel samples, showing that only metagenomic analysis could identify the pathogen. Herein, the absence of *E. coli* and *Salmonella* spp. was confirmed not only with conventional methods but also with NGS techniques, placing the sampling areas under study in class A (intended for direct human consumption).

Moreover, the bacterial abundance at the genus level within each region (between the different zones) of Chalastra and Makrigialos was investigated. Despite the fact that the hierarchy is seemingly resembling, the statistical analysis showed significant differences in the abundance of the respective genera between the zones of both regions. Minor hierarchical differences were also found in the study of Bozcal and Dagdeviren [2], where stations in close proximity were investigated. On the other hand, both hierarchical and abundance differences were found when comparing the two regions in winter and in summer time periods, as well as when comparing each region, separately, between winter and summer. Major alterations were noticed in *Anaplasma*, *Mycoplasma* and *Ruegeria* genera, probably due to the geographic distance of the two regions and to microenvironmental fluctuations, such as temperature and nutrient supply availability. Temperature displays a key role in the gut microbial community of *M. galloprovincialis*, as already indicated in Li et al. [37].

Regarding the microbiome of *Mytilus galloprovincialis*, Chalastra and Makrigialos regions have not been studied before, and as a result, these unmapped regions may contain unidentifiable bacterial strains, even in the most updated bioinformatic databases. The findings of our study, compared to others, point out that differences in the bacterial abundance

are more evident at lower taxonomic levels. These differences were found at all levels of comparison (seasons, regions and zones within each region) and may occur due to the dynamic character of the microorganisms, which is mainly affected by temporal and spatial changes, environmental fluctuations (e.g., pH, temperature, $O_2$ concentration, salinity) and anthropogenic factors (e.g., agricultural and industrial wastewater). However, it has not been fully investigated how the mussel's microbiome responds to these alterations. Thus, this pioneer study might set the baseline for future and more thorough analyses of *M. galloprovincialis* microbiota in Chalastra and Makrigialos culture areas.

## 5. Conclusions

The present research contributes to the characterization of *M. galloprovincialis* microbiome in areas that have not been studied yet. Till now, all relevant studies show comparable results in higher hierarchical taxa; however, there are noticeable differences at lower taxonomic levels such as family, genus and species. These differences might be a consequence of the dynamic nature of the microorganisms in general, the specific marine environment in which the mussel grows and/or being cultured and the genotyping methodology followed in each study for the analysis of the bacterial profile. An in-depth analysis of the microbiome is critical in order to clarify the role and function of the bacteria that coexist with *M. galloprovincialis*. This requires a more holistic approach with a greater number of samples and sampling areas, being analyzed in more time points throughout the year. The use of long-read Third Generation Sequencing technologies can greatly assist to that direction, providing a higher sequencing depth of the genes under study and, subsequently, a better classification resolution (down to the species level in some cases) along with updated bioinformatic tools and databases of microbial genomes.

**Supplementary Materials:** The following supporting information can be downloaded at: https://www.mdpi.com/article/10.3390/d15030463/s1, Figure S1: Comparison of bacterial diversity between Chalastra/Makrigialos sampling regions and winter/summer time periods at phylum and family taxonomic levels.

**Author Contributions:** Conceptualization, A.I. and D.L.; methodology, D.L. and D.C.; software, D.L., K.S., V.K., E.B. and G.F.; validation, D.L., A.I. and D.C.; formal analysis, D.L., K.S., V.K., E.B. and G.F.; investigation, D.L., K.S., V.K. and E.B.; resources, D.L., D.C., E.B. and G.F.; data curation, D.L. and D.C.; writing—original draft preparation, K.S. and V.K.; writing—review and editing, D.L. and A.I.; visualization, K.S. and V.K.; supervision, D.L. and A.I.; project administration, D.L. All authors have read and agreed to the published version of the manuscript.

**Funding:** This research received no external funding.

**Institutional Review Board Statement:** Not applicable.

**Data Availability Statement:** All data are available within the manuscript.

**Conflicts of Interest:** The authors declare no conflict of interest.

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
