# Peer review of "Microbiome Profile of the Mediterranean Mussel (Mytilus galloprovincialis) from Northern Aegean Sea (Greece) Culture Areas, Based on a 16S rRNA Next Generation Sequencing Approach"

_diversity, doi:10.3390/d15030463_

Round 1
Reviewer 1 Report
Manuscript by Schoinas K. et al. presents microbiome profile of the Mediterranean mussel. The paper is well written, organized and adds new understanding to the diversity of prokaryotes associated with mollusks.
Comments:
1. Is it really necessary to include "Third generation sequencing" in the title? In the manuscript itself, the authors successfully indicate the advantages and necessity of this method. This is quite enough. Moreover, there is no information about the differences in the results of analysis by the old and new methods on the example of the object under study.
2. The paper mentions the connection of the microbiota of bivalve mussels with seawater, but there is no data on the bacterial communities of the Northern Aegean Sea. The approach taken in the article on freshwater molluscs would certainly enhance the scientific work (https://doi.org/10.1016/j.scitotenv.2019.134915). It is enough even just to mention any information from literary sources about the water microbiome in those sites.
3. From the methodological part it is not entirely clear how many samples were analyzed? Tables 1 and 2 list 16 samples, but line 247 says “for all 15 samples”. Each sample has 10-12 individuals, which one has how many and what is the total number of individuals used?
4. Figure 2 shows the average of all 15/16 samples? How do the content of certain taxa differ at the level of phyla, classes, families in different zones, seasons? It is advisable to give a classic diagram as in the article https://doi.org/10.1371/journal.pone.0224796 (Fig. 2).
5. Has there been a deposit of data in public databases?
These comments would certainly strengthen the manuscript and make it more readable and useful for other scholars to use.
Author Response
We appreciate the constructive suggestions that the reviewers have made regarding our manuscript and we would like to thank them very much for their effort. We have made a number of significant changes, which have taken all the comments of the reviewers into consideration. We believe that these changes have substantially improved the manuscript in order to be acceptable for publication in Diversity journal. More specific comments on each of the points raised are the following:
Reviewer #1
- Is it really necessary to include "Third generation sequencing" in the title? In the manuscript itself, the authors successfully indicate the advantages and necessity of this method. This is quite enough. Moreover, there is no information about the differences in the results of analysis by the old and new methods on the example of the object under study.
‘Third generation sequencing’ in the title was replaced by ‘Next generation sequencing’.
Information regarding the results of bacterial analysis by using older methods (and the respective differences with newer methods) was added in the Discussion section (lines 352-363).
2. The paper mentions the connection of the microbiota of bivalve mussels with seawater, but there is no data on the bacterial communities of the Northern Aegean Sea. The approach taken in the article on freshwater molluscs would certainly enhance the scientific work (https://doi.org/10.1016/j.scitotenv.2019.134915). It is enough even just to mention any information from literary sources about the water microbiome in those sites.
Information from literary sources regarding the water microbiome in Thermaikos gulf was added in the Discussion section (lines 352-363).
3. From the methodological part it is not entirely clear how many samples were analyzed? Tables 1 and 2 list 16 samples, but line 247 says “for all 15 samples”. Each sample has 10-12 individuals, which one has how many and what is the total number of individuals used?
In total, 15 samples were analyzed. Tables 1 and 2 list 15 samples (M12 sample is missing).
The number of individuals (mussels) for each sample, as well as, the total number of individuals were added in Table 1.
4. Figure 2 shows the average of all 15/16 samples? How do the content of certain taxa differ at the level of phyla, classes, families in different zones, seasons? It is advisable to give a classic diagram as in the article https://doi.org/10.1371/journal.pone.0224796 (Fig. 2).
We have now analyzed the differences (and their corresponding significance as well) between the sampling regions and time seasons at the level of phylum and family. However, due to space restrictions, we have added the results (diagrams) of the analysis as supplementary material (Figure S1).
5. Has there been a deposit of data in public databases?
The genotypic data have not been deposited yet in a public database. The reason for that is because we are in the progress of analyzing the microbiome of extra mussel samples and water samples as well, for a shortcoming submission of a new research draft. Once we complete the ongoing analysis, we will deposit all genotypic data (old and new) in a database. Of course, the data from the present study are available if requested.
We hope that all of our corrections have met the referees’ comments. Please feel free to ask for any additional clarifications. Thank you for your time and effort.
Yours sincerely,
Dr. Dimitrios Loukovitis
Associate Professor
Department of Fisheries and Aquaculture
School of Agricultural Sciences
University of Patras
New buildings, PC. 30200, Mesolongi
Mobile: 6945418261
e-mail: dloukovi@upatras.gr; dloukovi@hotmail.com
Reviewer 2 Report
The submitted manuscript represents a contribution to the still necessary research on microbiome profile studies of the mussel genus Mytilus. The present research contributes to the characterization of M. galloprovincialis microbiome in areas that have not been studied previously.
The manuscript is clearly structured and informatively written. The procedure is described in a comprehensible way and the results are presented precisely.
Further comments:
L. 134/135: Why was the temperature not measured directly at the water depth where the mussels were collected? The significance would be much higher than just the temperatures at the water surface since the temperatures on the water surface can differ significantly from those in lower lying areas.
L. 150: The size of the mussels used is given very imprecisely. What size class was actually used?
L. 170: typo (elution)
L. 295: typo (1.9%)
Fig. 3 -5: please delete the digit after the decimal point because it is not relevant in this figure. The same applies for fig. 4 and 5. The statistical significances should also be shown in the figures 3-5.
Author Response
We appreciate the constructive suggestions that the reviewers have made regarding our manuscript and we would like to thank them very much for their effort. We have made a number of significant changes, which have taken all the comments of the reviewers into consideration. We believe that these changes have substantially improved the manuscript in order to be acceptable for publication in Diversity journal. More specific comments on each of the points raised are the following:
Reviewer 2:
1. L. 134/135: Why was the temperature not measured directly at the water depth where the mussels were collected? The significance would be much higher than just the temperatures at the water surface since the temperatures on the water surface can differ significantly from those in lower lying areas.
We totally agree with your suggestion. Indeed, the significance would be much higher if we had information of the temperature at the water depth where the mussels were collected. Unfortunately, at the time of sample (mussels) collection, we did not have the appropriate technical equipment in order to measure the water temperature at the specific depths. However, we have recently obtained the right technical gear and we are now in the progress of analyzing the microbiome of extra mussel samples and water samples as well (for a shortcoming submission of a new research draft), where we will include measurements of the water temperature at the exact collection points (depths).
2. L. 150: The size of the mussels used is given very imprecisely. What size class was actually used?
The size class was 6.5 - 7.5 cm. We corrected it in the manuscript (line 152)
3. L. 170: typo (elution)
Corrected
4. L. 295: typo (1.9%)
Corrected
5. Fig. 3 -5: please delete the digit after the decimal point because it is not relevant in this figure. The same applies for fig. 4 and 5. The statistical significances should also be shown in the figures 3-5.
Digits after decimal points were deleted. We have also marked the significances in Figures 3-5.
We hope that all of our corrections have met the referees’ comments. Please feel free to ask for any additional clarifications. Thank you for your time and effort.
Yours sincerely,
Dr. Dimitrios Loukovitis
Associate Professor
Department of Fisheries and Aquaculture
School of Agricultural Sciences
University of Patras
New buildings, PC. 30200, Mesolongi
Mobile: 6945418261
e-mail: dloukovi@upatras.gr; dloukovi@hotmail.com
Round 2
Reviewer 1 Report
Many thanks to the authors, they managed to make corrections. This improved the manuscript. I have no objection to its publication in Diversity.